# Direct Estimation of Equivalent Bioelectric Sources Based on Huygens’ Principle

**DOI:** 10.3390/bioengineering10091063

**Published:** 2023-09-09

**Authors:** Georgia Theodosiadou, Dimitrios G. Arnaoutoglou, Ioannis Nannis, Sotirios Katsimentes, Georgios Ch. Sirakoulis, George A. Kyriacou

**Affiliations:** Department of Electrical and Computer Engineering, Democritus University of Thrace, 67100 Xanthi, Greece; theod.georgia@gmail.com (G.T.); darnaout@ee.duth.gr (D.G.A.); ioannani@ee.duth.gr (I.N.); katsimentessoteres@gmail.com (S.K.); gsirak@ee.duth.gr (G.C.S.)

**Keywords:** electroencephalography (ECG), Huygens’ Principle, reciprocity theorem, finite element method

## Abstract

An estimation of the electric sources in the heart was conducted using a novel method, based on Huygens’ Principle, aiming at a direct estimation of equivalent bioelectric sources over the heart’s surface in real time. The main scope of this work was to establish a new, fast approach to the solution of the inverse electrocardiography problem. The study was based on recorded electrocardiograms (ECGs). Based on Huygens’ Principle, measurements obtained from the surfaceof a patient’s thorax were interpolated over the surface of the employed volume conductor model and considered as secondary Huygens’ sources. These sources, being non-zero only over the surface under study, were employed to determine the weighting factors of the eigenfunctions’ expansion, describing the generated voltage distribution over the whole conductor volume. With the availability of the potential distribution stemming from measurements, the electromagnetics reciprocity theorem is applied once again to yield the equivalent sources over the pericardium. The methodology is self-validated, since the surface potentials calculated from these equivalent sources are in very good agreement with ECG measurements. The ultimate aim of this effort is to create a tool providing the equivalent epicardial voltage or current sources in real time, i.e., during the ECG measurements with multiple electrodes.

## 1. Introduction

In an increasingly aging populace, a total global burden of disease has been raised [1] in the Western world. The leading contributors to the disease burden among the elderly are cardiovascular diseases (30.3% of the total burden in people aged 60 years and older), malignant neoplasms (15.1%), chronic respiratory diseases (9.5%), musculoskeletal diseases (7.5%), and neurological and mental disorders (6.6%) [1]. A substantial and increased proportion of morbidity and mortality due to chronic disease have already been observed in the elderly populace. This caused an increasing importance for more accurate and non-invasive methods for the diagnosis of such diseases. In addition, understanding the inner working mechanisms of the human body would play an important role in curing and treating them. These reasons have led to an increased demand for more accurate medical imaging techniques. The most well-known medical image methods include X-rays, computed tomography (CT) scans, magnetic resonance imaging (MRI), ultrasound, and nuclear medicine imaging, including positron-emission tomography (PET) [2]. Lately, techniques utilizing recordings of the electric activity of different parts of human bodies have emerged as novel medical imaging methods [3,4]. Electroencephalography (EEG) and electrocardiography (ECG) are the most widespread electric recording techniques, which monitor the electric activity of the brain and heart, respectively.

Electric recordings of living organs are utilized as a non-invasive method for detecting different anomalies of the living tissues and measuring the surface potentials of different parts of the body. For example, abnormal electric responses of different tissues (conductivity anomalies) may be used to indicate damage or the failure of a system before the first symptoms are displayed [3]. These passive techniques do not demand external electromagnetic sources that can harm the human body, such as CT and X-Rays [5,6]. Nowadays, the most used technique is the ECG to monitor the condition and detect different diseases of the heart. However, these recordings have not yet been fully understood, narrowing the possibility of a successful diagnosis [7]. For this reason, there is a great research effort towards transforming the conventional electrical recordings into an equivalent 2D or 3D representation of the tissue’s electric activity [8,9,10,11,12]. The ultimate aim of this work is the estimation of distributed equivalent voltage/current sources over the heart’s surface (epicardial) or the brain volume in real time. Estimating the equivalent sources permits the representation of the electric activity of the heart or brain in 3D targeting for the spatial detection of the different anomalies.

The potential to be able to characterize the electric activity of different organs using a non-invasive method for the computation of their surface potentials is a major breakthrough, especially in real-time applications (i.e., diagnosis of a cardiac arrest). Such non-invasive methods are commonly known as the inverse problem of electroencephalography [3,13,14] and electrocardiography [15,16,17], respectively, which have been studied to a great extent within the scientific community. Inverse problems aim at the specification of those equivalent sources responsible for the genesis of several known (measured) potentials on a surface that encloses them. In the case of the inverse electrocardiography problem, the known potentials correspond to the ECG recordings of a patient, and as for the equivalent source, it is the one representing the electric activity of the heart.

As referred to in [18], the inverse problem’s models in the literature include a single dipole with a time-varying unknown position; a fixed number of dipoles with fixed unknown positions; fixed known dipole positions; and a variable number of dipoles (i.e., a dipole at each grid point) but with a set of dipole moment constraints. As regards dipole moment constraints, which may be necessary to limit the search space for meaningful dipole sources, Rodriguez-Rivera et al. [19] proposed four dipoles’ models with different dipole moment constraints. These are (i) constant unknown dipole moment; (ii) fixed known dipole moment orientation and variable moment magnitude; (iii) fixed unknown dipole moment orientation and variable moment magnitude; (iv) variable dipole moment orientation and magnitude.

It is widely accepted in the literature related to biomedical inverse problems [20] that when the source location is sought, the problem becomes highly non-linear and thus very difficult to solve. On the contrary, when sources with fixed locations, with the unknown being their amplitudes and orientation (i.e., dipoles with unknown moments $P_{x}$, $P_{y}$, $P_{z}$), are considered, then the inverse problem becomes almost linear. This claim is essentially the key hidden behind the possibility of direct inverse problem solutions as carried out herein, without any need for iterative solutions. The effort within this work is directed to a different approach to inverse ECG problems. A distributed current source over the epicardium is sought for each temporal sampling. This approach serves the ultimate task of yielding a current and voltage distribution flowing over the cardiac muscle during its temporal cycle. In this manner, any damage in the cardiac muscle would be reflected in some irregularity in the epicardial current flow or voltage amplitudes. We performed a direct solution for either discrete (not presented herein) or distributed sources, as described next. In this case, the source origin (position) is assumed to be fixed (epicardial) but with an unknown magnitude (voltage or current). The methodology’s validation proved the claim of near linearity.

The first step in solving the classical inverse problem is the initial guess about the equivalent fixed location source within the thorax model. Having both the volume conductor model and the known sources, the solution of the forward problem takes place, resulting in a potential distribution throughout the model ($V_{c}$). For the inverse solution to be established, alongside the calculated potential distribution, a set of measured data needs to be acquired, denoted as ($V_{m}$). Within each iteration of the inverse problem, a cost function is established, which is usually defined as the sum of the squared differences (SSQ) between the calculated and the measured potentials [8,21]. The problem is iteratively solved until its residual is minimized. Within every iteration, the source parameters, i.e., the moments of the dipoles, are gradually altered, resulting in a different solution to the forward problem, which is newly compared to the measured potentials. Common search methods include the iterative Kozlov–Maz’ya–Fomin (KMF) method [10], genetic algorithms [22], the Newton–Raphson method, Gauss–Newton [23], the Levenberg–Marquardt algorithm [24], Tichonov regulation [11,25,26], and neural networks [12,27]. Nowadays, the main research effort is toward finding a more efficient regularization technique to increase the accuracy of the above algorithms [9,28,29,30,31,32,33]. For example, in [34], a “patchwork method”, which combines two classical numerical methods for inverse ECG—the method of fundamental solutions (MFS) and the finite-element method (FEM)—is studied to increase the accuracy of the reconstructing sinus rhythm, particularly breakthrough sites. Another field of research work is pointing to improving all existing regularization methods with alternative techniques, such as solving the problem in the wavelet domain, obtaining a sparse solution with multitask elastic net, and transforming it back to the time domain [35]. A few research efforts towards the estimation of the heart’s electric sources have been made that utilize machine learning algorithms, such as exploiting regression problems, constraining the solution space by decomposing signals with multidimensional Gaussian impulse basis functions [36], or utilizing Long Short-term Memory (LSTM) and Convolutional Neural Networks (CNNs) [12]. However, a major drawback of the inverse problems that are currently under use is the high computational time and resources needed. Indeed, in most works in the literature, the processing time is not even addressed when the inverse ECG mainly concerns the accuracy of the proposed model.

The method proposed within the context of this article addresses the above-mentioned problems, i.e., the decrease in the computational time and resources needed for the estimation of the equivalent brain or heart sources. The proposed method, based on Huygens’ Principle [37,38], aims at a direct estimation of equivalent bio-electric sources. The main difference between the classical approach of the inverse problem solution and the proposed method lies in the fact that the latter one does not require iterative solutions to the forward problem.

Specifically, using Huygens’ Principle, we can consider a closed surface (e.g., the thorax or head surface) surrounding the sources under study. Then, the potential values measured on those surfaces can be regarded as secondary Huygens’ sources, which, as can be theoretically proven, generate the same potential distribution as the primary ones [39]. In more detail, we initially consider an unknown function V(x,y,z) that generates the potentials within the volume-conductor model. The next step towards our final goal is to acquire a set of ECG or EEG measurements recorded from the surface of the thorax or the head, respectively. Using appropriate shape functions, we interpolate the data so as to acquire a potential surface distribution on the whole model surface to able to utilize Huygens’ Principle. Specifically, the secondary sources must have a non-zero value only on the selected closed surface, while they are zero on the rest of the volume model. In order to achieve a faster and computationally efficient method, the V(x,y,z) function is handled as an eigenfunctions’ expansion, as in our previous work [8]. Those eigenfunctions are extracted after the numerical solution of the eigenvalues problem, which is formulated using the finite elements method (FEM). We then equate the generated surface distribution with the expansion of the V(x,y,z) function, and by exploiting the orthogonality of the eigenfunctions and integrating over the thorax surface, we extract the weighting factors of the expansion. Knowing the weighting factors that correspond to our volume conductor model, we can consider that the function V(x,y,z) generates not only the surface potentials but also the potentials within the model, meaning that the heart’s surface potentials can also be estimated. Now is the time to apply Huygens’ Principle for the second time regarding the primary equivalent sources on the heart surface (epicardium). Like before, the sources are considered non-zero only on the heart’s surface, while they equal to zero anywhere else. By integrating the heart’s surface and using the FEM’s shape functions, we estimate the epicardium potentials.

An alternative approach refers to the estimation of equivalent current densities flowing over the epicardium. However, its explicit solution is left for future task. It is anticipated that the proposed method will enable an “epicardial source mapping” in almost real time. Namely, within a few minutes, the epicardial potential distribution could be extracted. Thus, if this is indeed possible, an important diagnostic tool could be established. Having the epicardial source distribution versus time, pathological cases can be distinguished with ease. Knowing the physiological activation timing for the heart, malfunctions of cardiac components, or even regions with no activation (i.e., necrosis), could be determined.

## 2. Materials and Methods

### 2.1. Volume Conductor Problem

For the solution of the studied biomedical cases, and any other problem in general, a proper model must be employed for the numerical solution to take place. Within the context of this article, the geometry of the thorax will be studied. The human body has a particularly complicated internal structure, as it comprises several different types of tissues that introduce a high inhomogeneity and must all be considered for the construction of a realistic volume conductor model. As expected, the higher the accuracy of the model, the higher the accuracy of the numerical method will be. Alongside the accuracy, however, the model is also closely correlated with the execution time of the algorithm. Thus, a trade-off between those two parameters must be considered for the models to be employed in real-time scenarios.

The volume conductor models used were taken from the classical related articles of Jorgenson [40], Gabriel [41] and the data base of ITIS [42] but in the form adapted from [43]. Specifically, Jorgenson et al. [40] suggested that a human torso needs more than 400,000 cubic elements to accurately model it. In order to utilize a model in this direct inverse problem, it is crucial to keep the number of elements as low as possible to minimize the computational time and resources maintaining the capability of real-time applications.

When talking about volume conductor models, we refer to discretization schemes that transform the continuous geometry to a finite number of elements that are homogeneous as units. The volume conductor model employed for the purposes of this article was in the context of the finite element method (FEM). One of the main advantages of the FEM method is that through the utilization of arbitrarily shaped elements (i.e., tetrahedrons, hexahedrons, etc.), it can describe almost any arbitrary given geometry. Depending on the problem, each element is characterized by properties that are related to the physical model, i.e., the conductivity or permittivity of the tissues. Since the complexity mostly lies in the curved boundaries between the tissues, the number of elements can be drastically reduced when arbitrarily shaped elements are used. For the current study, hexahedral volume elements are adopted from our previous works that displayed relatively good accuracy and low computational times [43,44].

The thorax model comprises 32 levels, or 33 sections [43]. The vertical distance between the sections is constant and equals to 1.27 cm (0.5″). Horizontally, and on average, the elements are about 1.34 cm, varying from 0.26 cm to 2.45 cm. In total, the model consists of 8800 general hexahedrons, where 9729 nodes are presented, from which 613 are surface nodes. Figure 1a illustrates a human thorax alongside the corresponding sections employed for the composition of the model, with the numbering on the left being the original one [45] and on the right the one employed herein.

The assembly of the thorax model, i.e., placing the sections above each other, yields the exploitation of a reference axis. This axis was located outside the model so as to not interfere with any of the sections. The distance between each section and the reference axis was not constant but was defined after considering the shape of the spine (Figure 1b). The above analysis resulted in the volume conductor model of Figure 1c, where the fixed-positioned heart elements are highlighted. It is understood that the heart moves in time and its geometry may vary between different persons. We are aware of related research on the dynamic modeling of the heart and we plan to try such modeling when the proposed methodology is mature enough. Elements consisting of more than one tissue are denoted with a combination of the corresponding letters. The conductivity $\sigma_{e}$ of such an element is calculated as
(1)$$\sigma_{e}=\frac{V_{1}\sigma_{1}+V_{2}\sigma_{2}+\cdots +V_{n}\sigma_{n}}{V_{e}},$$
where $V_{n}$ is the volume of tissue-n, $\sigma_{n}$ is the corresponding conductivity, and $V_{e}$ is the volume of the element.

### 2.2. Direct Inverse Problem Solution

In the proposed method, the term *inverse* does not describe an iterative solution of the forward problem, as it usually does, but it rather implies the direct extraction of an equivalent internal current density or voltage source distribution given a set of surface potential measurements. The methodology that follows is the basis for the proposed direct solution of the inverse problem. The methodology is organized in a step-by-step approach as follows.

Interpolate the discrete electrode voltages to generate equivalent surface potential source distribution.Consider an eigenfunction expansion for the volume potential distribution and estimate its weighing factors by equating it to the source distribution (of step 1) by exploiting the eigenfunction orthogonality.Assume a source distribution over the epicardium (or inside the brain) as an expansion of the FEM basis functions. Its weighing factors are estimated by equating to the volume potential and exploiting the FEM basis function’s orthogonality.The resulting internal sources are validated by comparing their generated potential to the original ECG or EEG measurements.

Let V(x,y,z) be the function that generates the potentials within and upon the thorax given a specified excitation and $V_{i}^{m}$, with i=1,2,…,M, the *M* measurements recorded from the thorax surface. These surface potentials are regarded as a secondary Huygens surface source. The equivalence principle was introduced by A. E. Schelkunoff in 1936 [46] and forms a more strict statement of Huygens’ Principle, which states that “each point of a wave front can be considered as the source of secondary wavelets that spread out in all directions with a speed equal to the speed of propagation of the waves” [47].

Given a source **g** and a surface **S** that encloses it, there is a certain source **h** distributed over the surface **S** that gives the same field outside (or inside) this surface. Thus, two source distributions that produce the same field within a region are said to be equivalent within that region. Consequently, when we are interested in the field of a specific region, prior knowledge of the actual sources is not needed as long as the equivalent sources can be sought.

Let us assume, for instance, that $J_{1}$ and $M_{1}$ are electric and a magnetic sources, respectively, located within the volume $V_{1}$ and generating the fields $\vec{H_{1}}$ and $\vec{E_{1}}$, as can be seen in Figure 2a. An equivalent problem to this could be the one presented in Figure 2b, where the internal sources have been removed and replaced with their equivalent surface sources $J_{s}$ and $M_{s}$ that generate the internal field, while the external field can be considered as null (Love’s Equivalence Principles). These equivalent surface sources are [39]
(2)$${\vec{J}}_{s}=\hat{n}\times \left({\vec{H}}_{1}\right)={\vec{J}}_{1}$$
(3)$${\vec{M}}_{s}=-\hat{n}\times \left({\vec{E}}_{1}\right)=-{\vec{M}}_{1}$$

In terms of the problem under study, **g** corresponds to the equivalent heart (or brain) activity sought, **S** corresponds to the thorax or scalp surface, and the auxiliary Huygens sources (**h**) can be extracted from the recorded surface potentials through interpolation (Figure 2c). As mentioned above, the surface distributed source **w** can produce the same field either inside or outside the enclosing surface **S**. We consider the case of electrostatic sources and electric fields, i.e., $\vec{M_{1}}=0$ and ∂/∂t=0, when the external field is considered null. The field equivalence principle has been exemplified for various different cases by Branko Popovich in his classic work [48]. Therein, the present case of the static or quasi-static problem is also analyzed. This distributed source can be either a current or a potential source. The problem formulation for both cases follows.

#### 2.2.1. Step 1: Surface Source Distribution

The equivalent surface sources are estimated from the measured potential (ECG or EEG) as
(4)$${\vec{J}}_{s}=-{\vec{J}}_{1}=-\left(\sigma {\vec{E}}_{1}\right)=-\sigma \left(-\nabla V_{1}\right)=\sigma \left(\nabla V_{1}\right)$$
(5)$${\vec{M}}_{s}=0$$
where $J_{s}$ the applied surface source, $J_{1}$ is the original one, $\vec{E_{1}}$ is the originally generated electric field, and $V_{1}$ is the voltage. In our case, $V_{1}$ corresponds to the measured potential over the surface of the model, i.e., $V^{m}$. However, to address Huygens’ Principle, distributed surface sources must be employed.

Explicitly, since not all surface thorax nodes correspond to a specific electrode, there are nodes for which there is no available information on measurements. Thus, to address this problem, the provided recorded potentials must be interpolated to result in a “measured” surface potential distribution that covers all nodes as seen in Figure 3.

A pyramid element interpolation of the provided voltage data was implemented to calculate the surface potential of the thorax/head [49]. Let us consider the pyramid element shown in Figure 4. In Figure 4, the purple circles suggest nodes for which the value is considered known, i.e., the potential measurement from the corresponding electrode, whereas the black ones denote nodes for which no information is available. In the problem under study, the electrodes are evenly placed throughout the thorax, having an electrode placed at every third level with a spacing of one node between every two electrodes of the same level, as can be seen in Figure 1c and Figure 4. To enable the calculation of the potential distribution, all pyramids were centered around nodes for which the potential was known. Thus, for the evaluation of each node, all the contributions of its surrounding electrodes must be considered. Roughly speaking, considering the interpolation shape function for the pyramid **$p_{k}$**, the potential for a specific node can be calculated as
(6)$$V_{i}=\sum_{j=1}^{M}\sum_{k=1}^{N}p_{k}V_{m,j}$$
where k=1,…,N, with *N* being the number of the pyramid nodes, and $V_{m,j}$ values are the measured potentials for all the j=1,…,M neighbor nodes that are associated with a specific electrode. The pyramid shape function was defined for a reference element with corners $A_{0},A_{1},A_{2},A_{3},A_{4}$ (Figure 4). The basis (shape) function for pyramids is not unique; thus, within the context of this article, the formulation presented in [49] was adapted, which approaches the pyramid formulation as the composition of two tetrahedrals, $P_{1}$ and $P_{2}$.

Having the recorded the potential on specific nodes, alongside the interpolation function, the potentials can now be evaluated, exploiting (6), over the whole surface of the thorax model, resulting in the anticipated surface potential distribution. In Figure 5, it is easily distinguishable that the pyramidal interpolation (red line) follows the original data (crosses) accurately for the 130^*th*^ time sample (130 ms), where the Q waveform is presented in this specific case. The x-axis displays the surface node numbering based on the thorax’s surface meshing of the FEM. The above procedure yields the potentials at all the time samples. However, the basic Huygens’ Principle asks for a continuous surface source distribution ($V_{d}$). This was obtained by exploiting the FEM interpolation-shape functions as implemented in [8], estimating every quadrilateral surface element surface potential ($V_{d}^{e}$):(7)$$V_{d}=\sum_{element}^{M}V_{d}^{e}\left(x_{s},y_{s},z_{s}\right)=\sum_{k=1}^{4}V_{k}^{e}N_{k}\left(x_{s},y_{s},z_{s}\right)$$
where *M* is the number of surface elements, $V_{k}^{e}$ is the estimated voltage of each surface element’s node from the pyramid interpolation scheme, and $N_{k}$ is the shape function implemented in the FEM. Having calculated the potential distribution, we can proceed to apply Hyugens’ Principle on the thorax’s surface.

#### 2.2.2. Step 2a: Volume Potential Eigenfunction Expansion—Current Sources

As explained in the previous subsection, the Huygens auxiliary source directly known from ECG recorded measurements is the voltage $V_{d}\left(x_{s},y_{s},z_{s}\right)$ over the thorax lateral surface. An equivalent auxiliary current source can be obtained through the conductivity of the tissue between successive surface nodes or the corresponding admittance of the FEM element. It is well established that each FEM element can be represented by an equivalent resistive (admittance *Y*) network. Considering two successive surface nodes *i* and *j* with distance $L_{ij}$ with estimated voltages $V_{d}^{i}$ and $V_{d}^{j}$, the current flowing on the branch $Y_{ij}$ can be approximated as
(8)$$I\left(x_{s},y_{s},z_{s}\right)=Y_{ij}\cdot \left(V_{d}^{j}-V_{d}^{i}\right)/\Delta L_{ij}$$

Let $I(x_{s},y_{s},z_{s})$ be the distributed current source over the surface of the model (thorax) and V(x,y,z) the generated potential throughout the volume conductor model. Therefore, the overall system can be described as
(9)$$I\left(x_{s},y_{s},z_{s}\right)=\nabla \left(\sigma \left(x,y,z\right)\cdot \nabla V\right)=\left[Y\right]V\left(x,y,z\right)=LV\left(x,y,z\right)$$
where L is the generalized Laplacian operator (for non-homogeneous media) and [Y] is the conductivity matrix of the FEM’s thorax model. The volume potential can be described as a modal eigenfunction expansion as presented in [50]:(10)$$\left[V\right]=\int_{V}w_{n}\left[{\hat{V}}_{n}\right]dV$$

The procedure of our previous work [8] is adopted to estimate the weighing factors $w_{n}$. Every pair of eigenvalue $\lambda_{n}$ and eigenvector ${\hat{V}}_{n}$ follows the property
(11)$$L{\hat{V}}_{n}=\left(\lambda -\lambda_{n}\right){\hat{V}}_{n}$$
where ${\hat{V}}_{n}$ values are the eigenvectors of the L Laplacian operator and $\lambda_{n}$ values are their corresponding eigenvalues. Equation (Equation 11) is multiplied by the weighing factors $w_{n}$, another eigenfunction (${\hat{V}}_{m}^{*}$), and then integrated over the whole domain volume as:(12)$$I=\int_{V}w_{n}L{\hat{V}}_{n}{\hat{V}}_{m}^{*}dV=\int_{V}w_{n}\left(\lambda -\lambda_{n}\right){\hat{V}}_{n}{\hat{V}}_{m}^{*}dV$$

Equation (Equation 12) establishes a relation with the distributed current source based on (9) and (10). Exploiting the orthogonality of the eigenvectors ${\hat{V}}_{m}^{*}$ and ${\hat{V}}_{n}$ ($\int_{V}{\hat{V}}_{n}{\hat{V}}_{m}^{*}=0$ for n≠m), we can observe that only when n=m will the integral be unity. The source is restricted only on the surface and is zero otherwise; thus, the right-hand-side integral is reduced to a surface one. The final relation is given as
(13)$$\int_{S}I\left(x_{s},y_{s},z_{s}\right){\hat{V}}_{m}^{*}dV=\left\{\begin{array}{cc} w_{n}\left(\lambda -\lambda_{n}\right) & for\;n=m\\ 0 & for\;n\neq m \end{array}\right.$$

Since the eigenfunctions ${\hat{V}}_{n}$ and ${\hat{V}}_{m}$ obey the orthogonality property (second side of (13)), the weighing factor is isolated as
(14)$$w_{m}=\frac{\int_{S}I\left(x,y,z\right){\hat{V}}_{m}^{*}dS}{\left(\lambda -\lambda_{m}\right)}$$

These weighting factors can be used to establish the eigenfunction expansion for the voltage throughout the torso volume.

#### 2.2.3. Step 2b: Volume Potential Eigenfunction Expansion—Voltage Sources

Equation (Equation 14) provides the volume potential distribution when a surface current $I(x_{s},y_{s},z_{s})$ is extracted. Alternatively, these can be estimated directly from the surface voltage $V_{d}\left(x_{s},y_{s},z_{s}\right)$ of Step 1 as follows. Considering a distributed voltage source $V_{d}\left(x_{s},y_{s},z_{s}\right)$ over the outer surface of the model $\left(x_{s},y_{s},z_{s}\right)$, we can assume that this is a subset of the function generating the voltage throughout the model volume V(x,y,z), i.e., $V(x_{s},y_{s},z_{s})$; thus,
(15)$$Outer\;Surface:\;\;V\left(x_{s},y_{s},z_{s}\right)=V_{d}\left(x_{s},y_{s},z_{s}\right)=\sum_{n}w_{n}{\hat{V}}_{n}$$

Herein, the desired potential distribution over the outside surface ($V_{d}$) is thought to be the interpolated voltage measurements recorded from the surface of the thorax or head, as discussed in Section 2.2.1. The general idea is to assume that the known voltage distribution $V_{d}$ can be generated from the same modal eigenfunction expansion of the whole model conductor. For the weighting factors to be calculated, the orthogonality of the eigenvectors is again exploited. Thus, multiplying (15) by an eigenvector ${\hat{V}}_{m}$ and exploiting the orthogonality property yields
(16)$$\int_{V}\sum_{n}w_{n}{\hat{V}}_{n}{\hat{V}}_{m}^{*}dV=\int_{V}V_{d}\left(x,y,z\right){\hat{V}}_{m}^{*}dV=\left\{\begin{array}{cc} w_{n} & for\;n=m\\ 0 & for\;n\neq m \end{array}\right.$$

Then,
(17)$$w_{m}=\int_{V}V_{d}\left(x,y,z\right){\hat{V}}_{m}^{*}dV$$

Nevertheless, keeping in mind Huygens’ Principle, the above integral degenerates to a surface one as the sources $V_{i}$ have values over the model surface and equal to zero elsewhere. Thus, Equation (Equation 17) reduces to
(18)$$w_{m}=\int_{S}V_{d}\left(x,y,z\right){\hat{V}}_{m}^{*}dS$$

#### 2.2.4. Step 3: Estimate the Internal Equivalent Sources

Knowing the weighting factors $w_{m}$, we can now consider that the same function (10) generates the potential distribution $V_{epic}\left(x,y,z\right)$ over the epicardium surface $S_{epic}$. The sought continuous epicardium voltage source is discretized with the aid of a finite element mesh as an expansion of the nodal voltage $V_{epic,k}$ values, where *k* is the epicardium nodes of the mesh. So we need to estimate these nodes’ potential to find the continuous voltage distribution, which reads as
(19)$$V_{epic}\left(x,y,z\right)=\sum_{k}V_{epic,k}N_{k}$$
where $N_{k}$ is the FEM interpolation shape function over the heart surface and $V_{epic}$ is the voltage on the surface of the epicardium. At this point, we employ Huygens’ Principle once again, only that this time, the equivalent sources are considered along the epicardium. The epicardium sources will be zero at every other point except from the epicardium surface. These epicardium voltages are assumed to be identical to the corresponding volume potentials obtained in step 2: (20)$$V\left(x,y,z\right)=\sum_{n}w_{n}{\hat{V}}_{n}=\sum_{k}V_{epic,k}N_{k}$$

Given the FEM’s shape functions $N_{k}\left(x,y,z\right)$ and their orthogonality properties, the following applies: (21)$$N_{k}\left(x,y,z\right)\cdot N_{m}\left(x,y,z\right)=\int_{V}N_{k}\left(x,y,z\right)N_{m}^{*}\left(x,y,z\right)dV=\left\{\begin{array}{cc} |N_{k}{|}^{2}, & \;k=m\\ 0, & k\neq m \end{array}\right.$$
where $|N_{k}{|}^{2}$ is 1 if the shape functions are normalized and is otherwise a normalization factor (nf). In turn, Equation (Equation 20) yields, when multiplied with another shape function $N_{m}^{*}\left(x,y,z\right)$,
(22)$$V\left(x,y,z\right)N_{m}^{*}=\sum_{n}w_{n}{\tilde{V}}_{n}N_{m}^{*}=\sum_{k}V_{epic,k}N_{k}N_{m}^{*}$$

Finally, by integrating over the heart surface (epicardium) and utilizing the (21), Equation (Equation 22) gives
(23)$$\int_{S_{epic}}\sum_{n}w_{n}{\tilde{V}}_{n}N_{m}^{*}dS=\int_{S_{epic}}\sum_{k}V_{epic,k}N_{k}N_{m}^{*}dS$$
(24)$$V_{epic,m}=\frac{\int_{S_{epic}}\sum_{n}w_{n}{\tilde{V}}_{n}N_{m}^{*}dS}{nf}$$

Having calculated the epicardium nodes’ potential, the continuous voltage distribution can be estimated using (18) over the epicardium surface. Summarizing, and in order to extract the epicardial potential distribution, the following 6 steps can be generally identified:Acquisition of a data set corresponding to measurements recorded from the surface of the thorax.Interpolation of the acquired recordings throughout the surface of the model, i.e., the thorax.Calculation of the weighting factor $w_{m}$, exploiting Equation (Equation 14) or (18), depending on the selected source type.Adaptation of the problem to describe the heart’s surface (Huygens’ Principle) and selection of the appropriate shape functions for them to comprise an orthogonal basis.Exploitation of the orthogonality over the heart and integration over the epicardium.Extraction of the epicardium potentials using Equation (Equation 24).

### 2.3. Numerical Implementation

The proposed method starts with the acquisition of a set of measured data. Within the proposed method, the measured data employed for the solution of the inverse problem are ECG recordings using a 192 electrode vest, and they are obtained from Utah University [51]. The volume conductor model, as explained, was implemented based on our previous works [8,43,44], in the commercial programming software Mathworks MatLab 2022b (Natick, MA, USA) [52], where all the processing steps were conducted.

Herein, the voltage source is thought to be the voltage measurements recorded from the surface of the thorax. This potential source distribution is non-zero only on the surface of the model and zero anywhere else. Having the potential distribution throughout the thorax calculated, alongside the eigenvectors resulting from the eigenanalysis of the conductivity matrix [Y], we can address our main objective. Thus, the weighting factors for each time instant can be extracted. The exploiting method in the current work uses a hybrid combination of Proper Orthogonal Decomposition (POD) and FEM on a torso model for rapid multiple solutions of the inverse problem that was proposed in our previous works [8,53]. The calculation of eigenvalues and eigenvectors for very high-rank matrices is a highly demanding task that requires a lot of computational resources as well as time. For this purpose, the whole eigenanalysis pre-processing was conducted overnight. After the eigenvalues and eigenvectors have been estimated and since the presence of any source does not alter the geometry or the electrical properties of the tissues, the potential generated can be easily represented with the expansion terms of (10). The last step is to calculate the weighting factors for every time step using (18).

As clearly suggested in (13), for the resulting weighting factors, the integration of the product $V_{d}{\hat{V}}_{m}^{*}$ over the thorax surface must be evaluated. An obvious solution to the problem would be the exploitation of Gauss integration through FEM over the corresponding quadrilaterals assembling the model’s surface. Such an approach, however, considering that the potential distribution refers to multiple time instances (consisting of 740 samples for each of the 128 selected electrodes), introduces an unacceptable computational burden, alienating us further from a fast solution to the inverse problem. Thus, to avoid this problem, the integral of Equation (Equation 18) was approximated from a different point of view. Let us consider two functions, $f_{1}$ and $f_{2}$, for which the following integral, with regards to Equation (Equation 17), is sought:(25)$$I_{int,i}={\;\int \int \;}_{S}f_{1}\left(x,y\right)f_{2i}\left(x,y\right)dxdy$$
where the subscript *i* of function $f_{2}$ denotes the *i*th time interval, as $f_{2}$ is time-dependent contrary to $f_{1}$. In our case, $f_{1}$ can be replaced with the extracted eigenvectors of the model, while $f_{2}$ stands in for the measured potential of the surface model’s nodes. Namely, we have

$f_{1}\to \left[V_{n}\right]$, which is an [n] array containing the *m* eigenvectors for each of the *n* nodes, and$f_{2}\to \left[V_{d}\right]$, which is an [n] array consisting of the *n* nodes of the model from which only the ones corresponding to the surface of the thorax are non-zero and the different time instances *t*.

The main objective is to associate the integral to the time-invariant functions, i.e., the eigenvectors, for it to be calculated only once. This is indeed a possible approximation as the time-varying function, i.e., the potential distribution, presents slight alterations between neighboring nodes, and thus, it can be considered as constant for a given quadrilateral for a time instance. The integral of the time-invariant function can be regarded as a geometrical factor that encapsulates all the information related to the geometry of the model’s surface.

The utilized approach is based on the exploitation of the isoparametric transformation employed in the context of FEM as proposed in [8]. As described above, considering the potential distribution $V_{d}^{i}$ constant over a quadrilateral for a time instance *i*, only the integral of the eigenvectors over the element needs to be evaluated. Namely,
(26)$$w_{m,1}=\int_{S}V_{i}{\hat{V}}_{m}^{*}dS=V_{d}^{i}\int_{S}{\hat{V}}_{m}^{*}dS,for\;i=1,2,\ldots ,N$$
where the *N* is the total number of samples, 740 in the current case. Denoting the result of the integration, obtained through FEM, that inherently contains the information about the model’s geometry as ${\left[V_{n}\right]}_{geom}$ and the samples of the potential distribution as ${\left[V_{d}\right]}_{i}$, Equation (Equation 22) can be expressed in its discrete form:(27)$$w_{i}={\left[V_{n}\right]}_{geom}^{T}\cdot {\left[V_{d}\right]}_{i}$$

Considering the extracted weighting factor, the epicardium’s potential distribution can now be calculated by utilizing Equation (Equation 24).

## 3. Numerical Results

The solution starts with the acquisition of an ECG potential distribution recorded from an adult’s thorax [51], which in our case was a healthy male. For the experimental setup and further information about the procedure of the measurements, refer to [51]. Given the net of electrodes employed during the recordings, the first step is to associate the electrode position with our volume conductor model’s node, i.e., the thorax model. This association is, of course, an approximation, and therefore, some errors are introduced to our computations. To avoid additional errors, not all recording electrodes were considered, but only 128 out of the 192 (12 levels × 16 electrodes) were used. Out of the 12 levels of electrodes in the original distribution, only the first 8 were selected, as can be seen in Figure 1c. Those four levels corresponded to the upper chest and were left out to minimize possible errors due to the shoulders of our original model. The data presents one cardiac circle or QRS complex where the measurement begins at 1 ms and ends at 740 ms, and the number of samples is 740, meaning that the sampling frequency is 1 kHz.

The inverse ECG problem starts with the solution of the system of Equation (8), as described in Section 2.2.2. For the evaluation of the problem, however, it is of crucial importance to inspect the eigenvalue distribution before we try to solve Equation (Equation 14) or (18). The model of thoraxes consists of 8800 hexahedral elements, based on the fact that the eigen-analysis will result in the same number of eigenvectors as the total unknown nodes (9729). The first step is to sort in ascending order the resulting eigenvalues as depicted in Figure 6a. As clearly presented in Figure 6a, the values of the first few eigenvalues are very small compared to the rest. If we zoom only in on the first 200 eigenvalues for visualization purposes, the corresponding distribution is given in Figure 6b, where the low eigenvalues are clearly distinguishable.

Note that the very small eigenvalues are usually highly inaccurate, and if we included them, they may cause instability or large errors in the expansion (10). Consequently, without affecting the accuracy of the calculated solution, it is possible to discard those very small eigenvalues to achieve better stability and a faster solution. The difficulty related to this approach is that, herein and in electromagnetics in general, the useful information lies in the small, low-order eigenvalues as depicted in Figure 7. Hence, the challenge is to discard only some very small eigenvalues that represent eigenvectors with dominant numerical “noise” inaccuracies. Since eigenvalues constitute the spectral representation of the structure, Parseval’s theorem applies. That is, the square of the weighing factors represents the power carried by that eigenmode. Considering the eigenvalues contributing to 99.9% of the total energy, 9589 eigenvalues out of 9729 are retained, as seen in Figure 6b. These 140 eigenvalues present low eigenvalues and weighing coefficients, so we can assume that they represent numerical noise, and they will degrade the efficiency of our algorithm if they are not removed. Even though this might seem like a small reduction in the matrix rank, considering the volume conductor models of higher discretization, the above-mentioned technique can result in an important decrease in the computational time without affecting the results’ accuracy. But most importantly, it highly improves the system matrix condition number, significantly improving the expansion accuracy; that is, the condition number is decreased from $7.8313\times 10^{5}$ to 96.4412. It is important to state that even after the rejection of just the first eigenvalue, the condition number decreases to 510.68, which suggests that the problem lies within the first few eigenvalues.

In order to interpret the results, it is necessary to represent all the heart components involved in cardiac activation. In Figure 8, the calculated potential distribution, as resulting from the direct inverse problem, is presented for several time instances. The calculated potential distribution over the epicardium tends to closely resemble the electric physiology of the heart, which is analyzed in detail in [54]. That is, starting at about 5 ms (with the signal origin being 1 ms), the area around the sinoatrial (SA) nodes, alongside some atrial nodes, is activated (10 ms), as in Figure 8a, when the P-waveform appears [54]. Some time instances later, i.e., in Figure 8b at t = 30 ms, the amplitude of the activation increases. The next nodes activated (Q-Waveform), around t = 130 ms, are the ones corresponding to the atrioventricular (AV) node, the Bundle of His, and the bundle branches (Figure 8c), following the anticipated cardiac activation. After this, the ventricular depolarization takes place, showing an R-waveform around 170 ms (Figure 8d), with the right ventricle activation slightly preceding it. After a while, i.e., around 200 ms, the S-waveform is displayed (Figure 8e), when the left ventricle depolarizes as well. At about 400 ms, the ventricular starts its repolarization (T-waveform), and all cells return to their resting potentials (Figure 8f).

## 4. Discussion

Considering an electrocardiogram of a healthy man, the above complies sufficiently with the physiological heart-activation sequencing [54]. The description in Figure 6.7 in [54] and its online version gives a detailed explanation of the action potentials generated by the specialized cells found in the heart, as well as how these are combined to yield the observed ECG recordings. This illustration also shows the timing of different parts activation (sinus node, atrial muscle, atrioventricular node, common bundle, bundle of branches, Purkinje fibers, and ventricular muscle). These time instances are selected so as to illustrate the reconstructed epicardial potentials in Figure 8 above. That is, the cardiac activation signal starts from the SA nodes, propagates through the atrial and through the AV node, the bundle of His, and the bundle branches, and the Purkinje fibres are transmitted to the ventricle. The main differences between the calculated results and the physiological heart activation are observed during the atrial depolarization, which in our case is not spread throughout the corresponding atrial nodes but is rather gathered around two distinct areas (Figure 8a,b). In addition to this, the AV node, the bundle of His, and the bundle branches tend to be activated almost simultaneously (Figure 8c), whereas their physiological activation presents a delay of several msec [54]. This may be a result of the poor discretization of the heart into elements, meaning that our mesh model needs to be modified to test this assumption.

Having the epicardium potential distribution (reconstructed sources), it is now possible to determine whether the extracted equivalent source can generate the anticipated potential distribution over the thorax. These are compared against the original ECG measurements for validation purposes. Figure 9 illustrates the generated potential upon the corresponding electrodes after the evaluation of the classical forward problem, given the resulting epicardial potentials. For the self-validation of the method, the extracted results are given alongside the original ECG measurements for four different electrodes positioned in different regions of the torso. Figure 9 corresponds to electrodes 9, 78, 114, and 141, which are located on the back, on the chest, on the right, and on the left side of the thorax’s patient, respectively. We randomly chose four electrodes to display their generated potentials, with the only limitation of belonging in different areas of the torso surface in order to highlight the different accuracy of the estimated sources’ resulting voltages as compared against recorded ECG measurements. The locations of the presented electrodes over the thorax model can be found in Figure 1c. The calculated results tend to have better behavior when the torso’s back and the chest are concerned, whereas the results regarding the sides of the torso are sometimes unable to sufficiently follow the original signal. The problem mainly lies in the simplistic volume conductor model, as the geometry of the body plays a crucial role in the extraction of accurate results based on the measurements. The next step will be to improve the model, keeping in mind the maintenance of the low processing time for real-time applications. For example, the voltage in electrode 114 (Figure 9c) fails to keep up with the high R waveform, and the voltage in electrode 141 (Figure 9d) presents the T waveform with a time delay with lower amplitude.

The proposed methodology requires significant computational resources and complex algorithms for the execution of the FEM analysis and the extraction of the eigenvectors utilizing the POD. However, this part is carried out in the pre-processing phase, i.e., the finite element modeling and the eigen-analysis. Thus, during the real-time source localization sought, the computational load is minimized, where the direct solution of the inverse problem is feasible through the proposed method within only 2 min, utilizing a Xeon E5 2650 v2 processor, whereas in the case of the classical inverse problem solution, the execution time could even exceed 2 h based on our work [50]. Unfortunately, to the best knowledge of the authors, the execution time is not referred to in any inverse ECG algorithms presented in the literature. Lacking this important metric, the value of our direct solution is not significantly highlighted. However, a comparison table is displayed (Table 1) where different inverse ECG methods are presented with their mean correlation coefficient (CC). At electrodes where the calculated results closely follow the original data, i.e., Figure 9b, the correlation coefficient has been estimated as 92%. There are, however, cases where this percentage significantly decreases, as is obvious in Figure 9c. In general, this deviation varies between 5 and 15% depending on the electrode position leading to a mean CC of 84%. Based on Table 1, our proposed method has equivalent accuracy with respect to other utilized methods in the literature. Furthermore, it seems that our method sufficiently estimates the results utilizing a smaller number of electrodes except in the cases of the [16,35], where they utilized a custom-generated human torso model obtained from magnetic resonance imaging (MRI) with high reproduction accuracy.

The most important outcome of the epicardial source distribution obtained is the prospect of an “epicardial source mapping”. That is, if it is possible to map the epicardial sources versus time, then an important diagnostic tool could be established for real-time applications, where the diagnosis must be made in minutes (e.g., cardiac arrests). The prospect is that for a physiological cardiac operation, the whole epicardial surface will be activated according to the proper timing. On the contrary, pathological heart functioning may result in either wrong timing in the activation of the heart components or no activation at all (e.g., partial necrosis). Hence, the proposed methodology can be exploited as an advanced diagnosis tool that in a very fast and convenient way takes into account ECG (or EEG) measurements over the entire thorax (or scalp) surface.

## 5. Conclusions

The main scope of this work was to establish a new, fast approach to the solution of the inverse electrocardiography problem. Based on Huygens’ Principle, measurements obtained from a patient’s thorax surface were interpolated over the surface of the employed volume conductor model and considered as a secondary Huygens’ source. This source, being non-zero only over the surface under study, was employed to determine the weighting factors of the eigenfunctions’ expansion describing the generated voltage distribution over the whole conductor volume, including the heart. For the accurate determination of the epicardial potential distribution, an integration over both the thorax and the heart’s surface is needed. The extracted results not only sufficiently estimated the heart’s potential distribution as denoted in Table 1 with other similar works, but they were also obtained within about 2 min, decreasing the execution time remarkably regarding the classical inverse problem formulation where several hours were needed for the solution to the problem. Having the ability to accurately extract the potential distribution over the epicardium within such a small amount of time, a significant diagnostic tool can be established, enabling the epicardial source mapping. That is, given the normal heart activation timing, different pathological heart conditions can be distinguished by observing possible alteration in the activation timing or even no activation at all (i.e., partial necrosis).

## 6. Future Extensions

The measured methodology may be extended in several directions with the ultimate task of achieving a real-time inverse problem solution. The first priority is to solve the ECG inverse problem for the current density sources following the exact reciprocity principles as in [48]. This is expected to improve accuracy but also to more accurately reflect the epicardial electrical functioning through the current density flow. The second priority is to improve the volume conductor model of Figure 1, primary by increasing the surface mesh density, in order to allow all measuring electrodes to be mapped on the model. This will allow the exploitation of all available measurements. The third priority is to improve the algorithm speed by employing state-of-the-art linear algebra techniques. Finally, the resulting direct inverse problem solutions will be applied to electroencephalography, i.e., to estimate equivalent sources for epileptic regions or to study the activated brain areas during challenging phenomena such as the default brain mode.

## Figures and Tables

**Figure 1 bioengineering-10-01063-f001:**
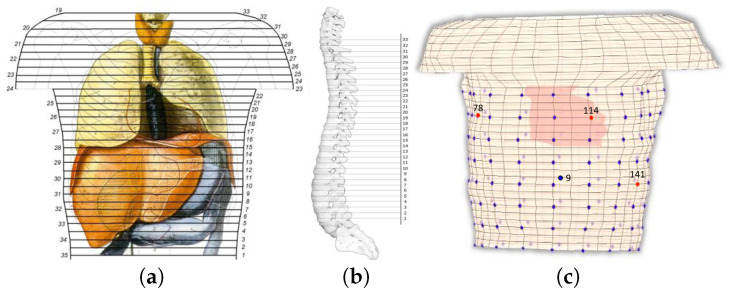
(**a**) Vertical view of the human torso, where the basic organs are presented alongside the horizontal sections of the anatomic atlas [45]. The numbering on the left corresponds to the original sections, whereas on the right is the employed numbering for the model. (**b**) Side view of the spine, where the reference axis and the model’s sections are also illustrated. (**c**) Discretized thorax model for FEM simulation, where the cardiac elements and electrodes position are also illustrated, the red dots are the electrodes (78, 114, and 141) in the front (thorax), and the blue dot is the electrode 9 in the back.

**Figure 2 bioengineering-10-01063-f002:**
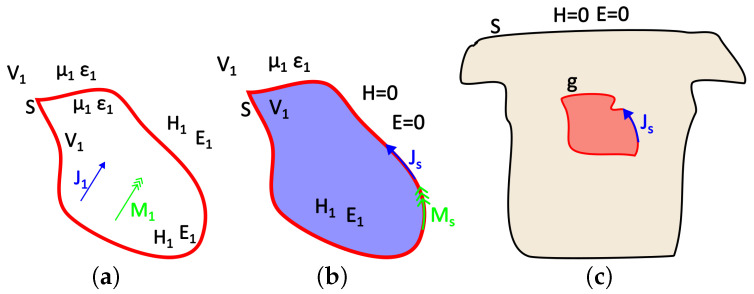
Illustration of (**a**) the actual model, (**b**) Love’s equivalent internal model, and (**c**) the equivalent problem of the thorax and epicardium currents.

**Figure 3 bioengineering-10-01063-f003:**
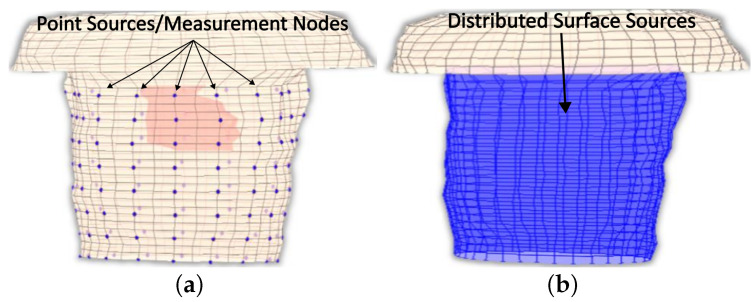
The employed thorax model, where (**a**) the different recorded sites and (**b**) the anticipated surface potential distribution are presented.

**Figure 4 bioengineering-10-01063-f004:**
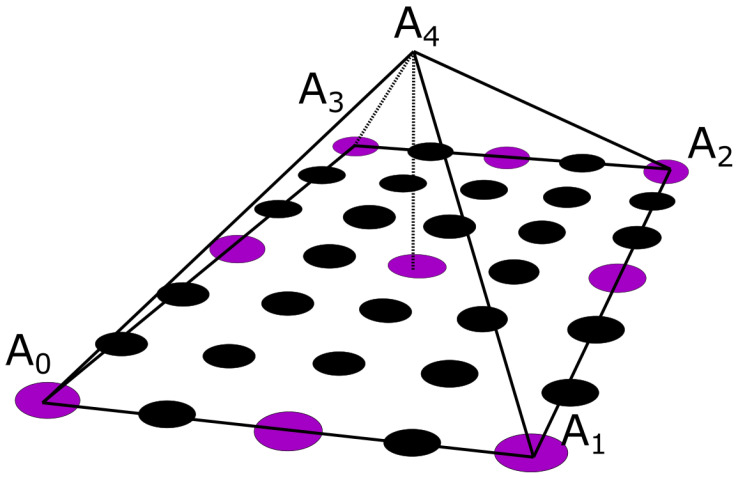
The pyramidal interpolation function as denoted upon the utilized node configuration. The voltage is measured on purple nodes and is additionally sought and interpolated on black nodes.

**Figure 5 bioengineering-10-01063-f005:**
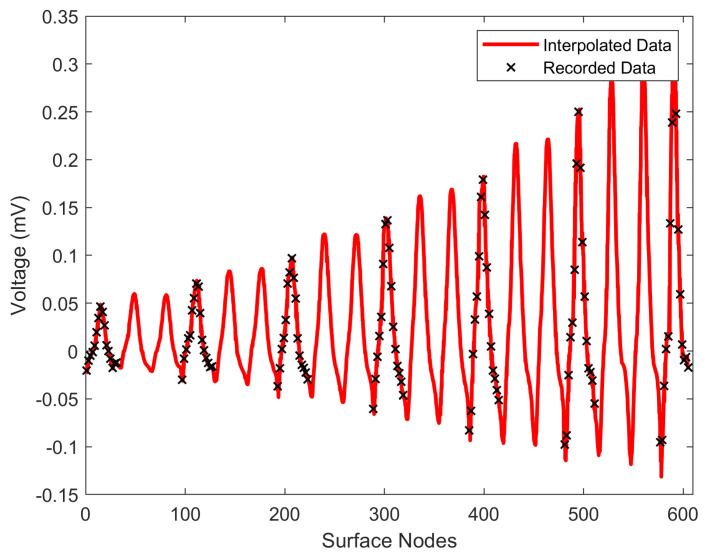
Comparison between pyramid interpolation (red line) and the original measurements (black crosses) for the surface nodes at 130 ms.

**Figure 6 bioengineering-10-01063-f006:**
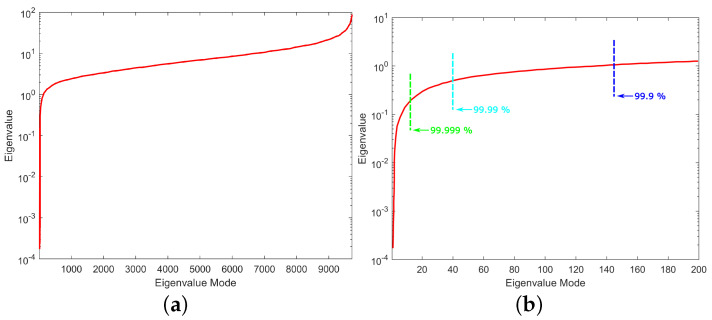
Distribution of eigenvalues for (**a**) the whole range and (**b**) the first 200 eigenvalues with the different energy percentages considered.

**Figure 7 bioengineering-10-01063-f007:**
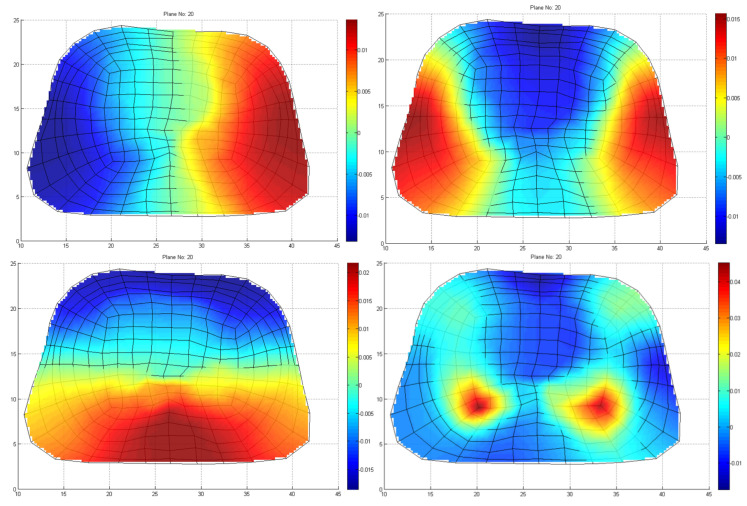
Four indicative low–order eigenvectors of thoraxes at the 20th cross–section, including a slice of heart, as depicted in Figure 1.

**Figure 8 bioengineering-10-01063-f008:**
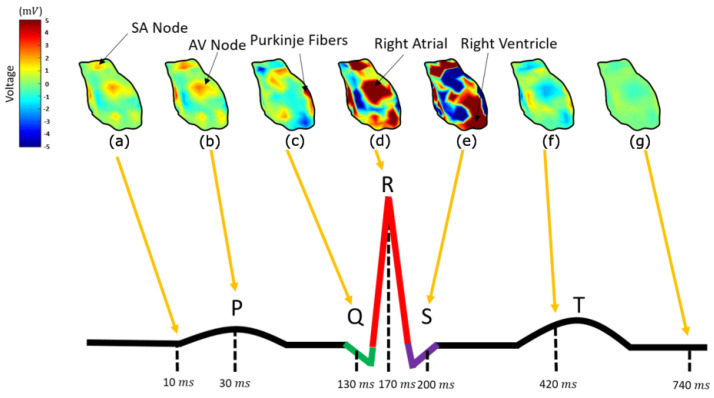
The epicardium potential distribution resulting from the inverse direct method for t=8 ms (**a**), t=30 ms (**b**), t=130 ms (**c**), t=170 ms (**d**), t=200 ms (**e**), t=420 ms (**f**), and t=740 ms (**g**) compared to a physiological PQRST electrocardiograph.

**Figure 9 bioengineering-10-01063-f009:**
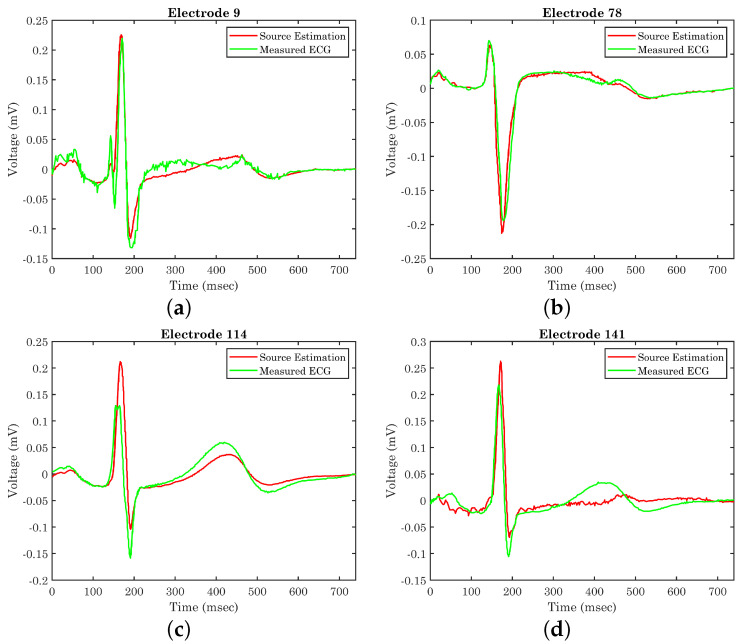
Original ECG measurements versus generated potential on electrodes (**a**) 9, (**b**) 78, (**c**) 114, and (**d**) 141 after the evaluation of the resulted equivalent source.

**Table 1 bioengineering-10-01063-t001:** Comparison table of correlation coefficient (CCs) for different methods in the literature.

Implemented Method	Electrodes	Mean CC	Reference
**Conventional ECG**	1002	90%	[55]
**Patchwork Method**	252	70%	[34]
**Tikhonov 0th order**	12	80%	[56]
**CARPentry**	12	89%	[16]
**Wavelet/Elastic Net**	192	79%	[35]
**CNN**	239	74%	[12]
**Proposed method**	128	84%	this work

## Data Availability

Not applicable.

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
