# Peer review of "Direct Estimation of Equivalent Bioelectric Sources Based on Huygens’ Principle"

_bioengineering, 2023, doi:10.3390/bioengineering10091063_

Round 1

Reviewer 1 Report (Previous Reviewer 2)

I am happy with the revised version. Authors tried to address all the comments. 

Reviewer 2 Report (Previous Reviewer 3)

thank you for the answers and the modifications.

Improved.

Reviewer 3 Report (Previous Reviewer 4)

The authors have responded to all my questions and queries. I am satisfied with the corrections made to the manuscript. 

The authors have addressed most of the typos. I suggest the authors to have their final manuscript read by a native English speaker before publication. 

This manuscript is a resubmission of an earlier submission. The following is a list of the peer review reports and author responses from that submission.

Round 1

Reviewer 1 Report

The aim of this effort is to estimate the equivalent epicardial voltage or current sources in real time.

Why no T waveform in Figure 8d 141 after the evaluation of the resulted equivalent source? But the measured ECG has T waveform.

Discussion and conclusions are very simple.

Moderate editing of English language required.

Reviewer 2 Report

Summary:

The paper describes a novel method for estimating electric sources in the heart using Huygens' Principle. The goal of this method is to solve the inverse electrocardiography problem quickly and directly. The study used recorded electrocardiograms (ECGs) and interpolated measurements from the patient's thorax surface as secondary Huygens' sources. These sources were used to determine weighting factors for eigen-functions that describe the voltage distribution in the volume conductor model. By applying the reciprocity theorem, equivalent sources over the pericardium were obtained. The methodology was validated by comparing the calculated surface potentials from these sources with ECG measurements. The ultimate aim is to develop a real-time tool that provides the equivalent epicardial voltage or current sources during ECG measurements with multiple electrodes.

Strength:

The Huygens Principle allows for precise localization of bioelectric sources by considering the wavefront propagation and the interaction with surrounding tissues or media. This can provide detailed information about the location of bioelectric sources in biological systems.

If the technique based on Huygens Principle is non-invasive, it can offer the advantage of not requiring direct contact or invasive procedures, which makes it safer and more acceptable for studying bioelectric activity in living organisms.

Depending on the implementation, Huygens Principle-based methods can offer computational efficiency, enabling the estimation of equivalent bioelectric sources in real-time or near real-time scenarios.

Limitations:

Any method based on Huygens Principle may rely on certain assumptions and simplifications about the underlying biological system, such as assuming a homogeneous medium or neglecting complex interactions. These assumptions can limit the accuracy or applicability of the technique in real-world scenarios.

Bioelectric measurements are often susceptible to noise and artifacts from various sources, including environmental interference or physiological factors. Huygens Principle-based methods may be sensitive to such noise and artifacts, which can affect the accuracy of source estimation.

While computational efficiency can be an advantage, some implementations of Huygens Principle-based methods may require significant computational resources or complex algorithms. This can limit the real-time applicability or practicality of the technique, especially for large-scale systems or high-resolution imaging.

Weakness of the article:

·       Experimental set up is not clearly described.

·       The literature section is weak.

·       Methodology needs to be significantly improved.

·       Without proper explanation the results are not clear and suspicious. 

Reviewer 3 Report

In this manuscript, the authors utilized a novel method based on Huygens' Principle to estimate electric sources in the heart from ECG measurements, aiming to solve the inverse electrocardiography problem quickly. Electrocardiograms were recorded, and measurements from a patient's thorax surface were interpolated over a volume conductor model to find secondary Huygens' sources. These sources were then used to determine the voltage distribution, allowing the calculation of equivalent sources over the pericardium. The method was self-validated and aims to provide real-time epicardial voltage or current sources during ECG measurements with multiple electrodes.

The idea behind this work is very promising. However, the manuscript is challenging to follow. Several parts need revision, along with the work plan and validation.

  1. Between lines 38 and 61, there are numerous claims without references. Adding references would be beneficial.

  2. Lines 107-144, explaining the main idea of the work, are confusing and hard to follow.

  3. Eq 10 appears incorrect as conductivity is missing.

  4. Part 2.1 is unclear and difficult to follow.

  5. There may be a missing integral sign in eq 22.

  6. The aspect of interpolating measured voltages to the thorax surface is not discussed. Combining this step with the main algorithm, as both are FEM models, could be helpful.

  7. The mathematical part is challenging to follow, with repetitions and unclear explanations.

  8. While the heart surface is expected to be known for this study, the authors should comment on its usual absence in similar cases.

  9. Computational details, such as the program used for meshing the problem or the approach, are missing.

  10. The self-validation of the data is weak. Starting with the direct model (voltage sources on the heart surface) and discussing simulated ECG values, interpolation, errors, eigenvalues, etc., would strengthen the validation process before addressing the inverse problem.

  11. An attached file highlights several typos and unclear sentences.

The manuscript is not clear and it is difficult to follow. Some typos are also present.

Reviewer 4 Report

This paper reports a direct estimation of equivalent bioelectric sources based on the Huygens principle. I have the following comments/concerns about the submitted manuscript that could improve its overall quality and readability.

1.       The authors need to provide the main contribution and novelty of their proposed technique over the already reported in the literature. I suggest the authors include a table of comparison for this purpose where the main parameters of the proposed technique are compared against those reported in the literature.

2.       In section 2, Materials and Methods, the authors discussed the trade-off between the execution time of the algorithm and the accuracy of the model. How did the authors make sure that the trade-off is taken care of?  

3.       Is there any specific reason for choosing 9729 nodes in the torso, and how are they counted?

4.       In section 2.3, Numerical Implementation, the approach utilized is based on the exploitation of the isoparametric transformation in the context of FEM. This approach should be discussed in detail.

5.       In Fig. 6, why only the first 200 eigenmodes are plotted for analysis?

6.       In Fig. 7, the epicardium potential distribution is shown from the inverse direct method. The figure is not very well explained in the text. For example, What is an SA node? What do S and A stand for? The authors are expected to provide some references, as to why the time scales of milliseconds were chosen to capture the cardiac activity. What is so special about 10 ms, 30 ms, and especially between 130 -200 ms?  The authors should also comment on, why the cardiac activity presents a delay of several milliseconds with references from the literature.

7.       In Fig. 8, why the specific electrodes are chosen for generated potential measurements?  

8.       Section 4. i.e., the discussion section; should be elaborated and the comparison with other related works in a tabular form could be included here.

There are many typos in the manuscript that need to be fixed. Some instances are as follows; page 1, line 20, elders should be replaced with elderly, line 25, demand of should be replaced with demand for, line 27, a important should be replaced with an important, line 28, have lead to should be replaced with have led. Page. 2, line 43, fully understand should be replaced with fully understood. Page 3, line 110, the word principal should be replaced with principle, line 116, theoretical proven should be replaced with theoretically proven. Page 5, line 196, the word were should be replaced with where. Page 6, the word principal should be replaced with principle. Page 8, the word “dissertation” should be replaced with “article".

There are many typos in the manuscript that need to be fixed. Some instances are as follows; page 1, line 20, elders should be replaced with elderly, line 25, demand of should be replaced with demand for, line 27, a important should be replaced with an important, line 28, have lead to should be replaced with have led. Page. 2, line 43, fully understand should be replaced with fully understood. Page 3, line 110, the word principal should be replaced with principle, line 116, theoretical proven should be replaced with theoretically proven. Page 5, line 196, the word were should be replaced with where. Page 6, the word principal should be replaced with principle. Page 8, the word “dissertation” should be replaced with “article".